

# A multi-stage heuristic method for service caching and task offloading to improve the cooperation between edge and cloud computing

Xiaoqian Chen[1], Tieliang Gao[2], Hui Gao[1], Baoju Liu[3], Ming Chen[4] and Bo Wang[4]

[1] Management Center of Informatization, Xinxiang University, Xinxiang, China
[2] Key Laboratory of Data Analysis and Financial Risk Prediction, Xinxiang University, Xinxiang, China
[3] School of Information Engineering, Pingdingshan University, Pingdingshan, China
[4] Software Engineering College, Zhengzhou University of Light Industry, Zhengzhou, China

## ABSTRACT

Edge-cloud computing has attracted increasing attention recently due to its efficiency on providing services for not only delay-sensitive applications but also resource-intensive requests, by combining low-latency edge resources and abundant cloud resources. A carefully designed strategy of service caching and task offloading helps to improve the user satisfaction and the resource efficiency. Thus, in this article, we focus on joint service caching and task offloading problem in edge-cloud computing environments, to improve the cooperation between edge and cloud resources. First, we formulated the problem into a mix-integer nonlinear programming, which is proofed as NP-hard. Then, we proposed a three-stage heuristic method for solving the problem in polynomial time. In the first stages, our method tried to make full use of abundant cloud resources by pre-offloading as many tasks as possible to the cloud. Our method aimed at making full use of low-latency edge resources by offloading remaining tasks and caching corresponding services on edge resources. In the last stage, our method focused on improving the performance of tasks offloaded to the cloud, by re-offloading some tasks from cloud resources to edge resources. The performance of our method was evaluated by extensive simulated experiments. The results show that our method has up to 155%, 56.1%, and 155% better performance in user satisfaction, resource efficiency, and processing efficiency, respectively, compared with several classical and state-of-the-art task scheduling methods.

# INTRODUCTION

As the development of computer and network technologies, mobile and Internet-of-Thing (IoT) devices have become more and more popular. As shown in the latest Cisco Annual Internet Report (*Cisco, 2020*), global mobile and IoT devices will reach 13.1 billion and 14.7 billion by 2023, respectively. However, a mobile or IoT device have very limited computing and energy capacities, constrained by the limited size (*Wu et al., 2019*).

Corresponding authors
Tieliang Gao, chrisgtl2012@163.com
Bo Wang, wangb@zzuli.edu.cn

Thus, user requirements usually cannot be satisfied by only using device resources, as mobile and IoT applications are the rapid growing in all of number, category and complexity. Globally, 299.1 billion mobile applications will be downloaded by 2023 (*Cisco, 2020*).

To address the above issue, mobile cloud computing is proposed to expand the computing capacity by infinite cloud resources (*Rahmani et al., 2021*). However, cloud computing generally has a poor network performance, because it provides services over the Internet. This leads to dissatisfactions of demands for latency-sensitive applications. Therefore, by putting a few computing resources close to user devices, edge computing is an effective approach to complement the cloud computing (*Huda & Moh, 2022*). Edge computing can provide services with a small network delay for users, because it usually has local area network (LAN) connections with user devices, over such as WIFI, micro base stations.

Unfortunately, an edge computing center (edge for short) generally is equipped with only a few servers due to the limited space (*Wang et al., 2020b*). Thus, edges cannot provide all services at the same time, due to their insufficient storage resources. In an edge-cloud computing, a request can be processed by an edge only when its service is cached in the edge. To provide services efficiently by edge-cloud computing, the service provider must design the service caching and the task offloading strategies carefully (*Luo et al., 2021*). The service caching decides which services are cached on each edge, and the task offloading decides where is each request task processed. There are several works focusing on both service caching and task offloading problems. However, these works have some issues must be addressed before their practical usages. Such as, several works assume there is an infinite number of communication channels for each edge, which ignored the allocation of edge network resources; (*Xia et al., 2021a*, *2021c*; *Zhang, Wei & Jin, 2021b*). Some works focus on the homogeneous requests (*Farhadi et al., 2021*, *2019*), which have limited application scope. All of the existing related works use edge resources first for a better data transfer performance, and employ cloud resources only when edge resources are exhausted. This can lead to an inadequate usage of abundant cloud resources. By using these existing service caching and task offloading strategies, some requests that can tolerate a certain latency are offloaded to edges at first. This can result in insufficient edge resources for meeting low-latency requirements of some subsequent requests.

To address issues of existing works, in this article, we focus on the joint service caching and task offloading problem for improving the cooperation between edge and cloud computing. We first formulate this problem into a mix-integer non-linear programming (MINLP) for optimizing the user satisfaction and the resource utilization. In order to solve the problem with polynomial time, we proposed a multi-stage heuristic method. The aim of our method is to make full use of both the low-latency of edge resources and the abundance of cloud resources. In brief, the contributions of this article are as follows.

- We formulated the joint service caching and task offloading problem into a MINLP for edge-cloud computing with two optimization objectives. The major one is to maximize

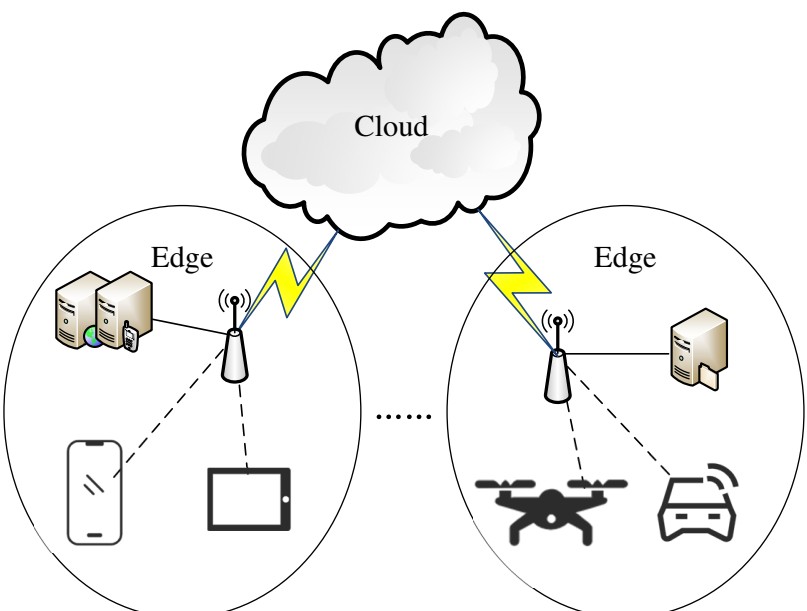

**Figure 1 The edge-cloud computing system.**

the user satisfaction in terms of the number of tasks whose requirements are satisfied. The minor one is maximizing the overall resource utilization.

- We proposed a heuristic method with three stages to address the joint service caching and task offloading problem in polynomial time. In the first stage, the method pre-offloads latency-insensitive request tasks to the cloud for exploiting the abundance of cloud resources. At the second stage, the proposed method processes latency-sensitive requests in the edge by caching their requested services on edge servers. In the last stage, our method re-offloads some requests from the cloud to the edge for improving their performance, when there are available edge resources at the end of the second stage.

- We conducted extensive simulated experiments to evaluate the performance of our proposed heuristic method. Experiment results show that our method can achieve better performance in user satisfaction, resource efficiency, and processing efficiency, compared with five of classical and state-of-the-art methods.

In the rest of this article, The next section formulates the joint service caching and task offloading problem we concerned. The third section presents the proposed multi-stage heuristic method. The fourth section evaluates the proposed heuristic approach by simulated experiments. The subsequent section illustrates related works and the last section concludes this article.

## PROBLEM FORMULATION

In this article, as shown in Fig. 1, we consider the edge-cloud computing system consisting of various user devices, multiple edges and one cloud. Each device has a wireless connection with an edge over various access points, and has a wide area network (WAN) connection with the cloud. Each edge is equipped with one or more edge servers. For

**Table 1  Notation description**

| Notation | Description |
|---|---|
| $T$ | The number of tasks. |
| $t_i$ | The $i^{th}$ task. |
| $f_i$ | The size of the computing resources required by $t_i$. |
| $a_i$ | The amount of the input data of $t_i$. |
| $d_i$ | The deadline of $t_i$. |
| $p_i$ | The start time of the input data transfer for $t_i$ that offloaded to an edge server. |
| $q_i$ | The finish time of the input data transfer for $t_i$ that offloaded to an edge server. |
| $c_i$ | The start time of the computing for $t_i$ that offloaded to an edge server. |
| $z_i$ | The finish time of the computing for $t_i$ that offloaded to an edge server. |
| $p_i^C$ | The start time of the input data transfer for $t_i$ that offloaded to a cloud server. |
| $q_i^C$ | The finish time of the input data transfer for $t_i$ that offloaded to a cloud server. |
| $c_i^C$ | The start time of the computing for $t_i$ that offloaded to a cloud server. |
| $z_i^C$ | The finish time of the computing for $t_i$ that offloaded to a cloud server. |
| $E$ | The number of edge servers. |
| $e_j$ | The $j^{th}$ edge server |
| $b_j$ | The storage capacity of $e_j$. |
| $g_j$ | The computing capacity of $e_j$. |
| $N_j$ | The number of communication channels provided by $e_j$. |
| $w_j$ | The communication capacity each channel in $e_j$. |
| $V$ | The number of cloud servers. |
| $v_k$ | The $k^{th}$ cloud server. |
| $g_k^v$ | The computing capacity of $v_k$. |
| $w^C$ | The network capacity of each cloud server. |
| $S$ | The number of services. |
| $s_l$ | The $l^{th}$ service. |
| $h_l$ | The storage space required by $s_l$. |
| $o_{i,j}$ | The binary constant indicating whether $t_i$ can be offloaded to $e_j$. |
| $r_{i,l}$ | The binary constant indicating whether $t_i$ requests $s_l$. |
| $x_{i,j,m}$ | The binary variable indicating whether $t_i$ is offloaded to $e_j$ and $m^{th}$ channel of $e_j$ is allocated to $t_i$ for the data transmission. |
| $x_{i,j}^e$ | The binary variable indicating whether $t_i$ is offloaded to $e_j$. $x_{i,j}^e = \sum_{m=1}^{N_j} x_{i,j,m}$. |
| $y_{i,k}$ | The binary variable indicating whether $t_i$ is offloaded to $v_k$. |
| $N_{fin}$ | The number of tasks with deadline satisfactions. |
| $U$ | The overall computing resource utilization. |

each request task launched by a device, it can be offloaded to an edge server (ES) or the cloud. When a task is offloaded to an ES, the ES must have a network connection with its device[1], and its requested service must be cached on the ES. If the task is offloaded to the cloud, there must be a cloud server (CS) that can meet all of its requirements. Next, we present the formulation for the joint service caching and task offloading problem in detail. The notations used in our formulation are shown in Table 1.

[1] The device of a task means the device that launches the request task.

## System model

In the edge-cloud computing system, there are $E$ ESs respectively represented by $e_j, 1 \leq j \leq E$. The storage and computing capacities of edge server $e_j$ respectively are $b_j$ and $g_j$. We assume the communications between user devices and ESs employ the orthogonal frequency multi-access technology, as done by many published works, *e.g.*, (*Tian et al., 2021*; *Wu et al., 2021*). For $e_j$, there are $N_j$ communication channels for the data transmission of offloaded tasks. As the result data is much less than the input data (*Gu & Zhang, 2021*; *Peng et al., 2021*; *Zhao, Lu & Chen, 2021*), we ignored the communication latency caused by the output data transmission. The communication channel capacity can be easily achieved by the transmission power, the channel bandwidth and the white Gaussian noise, according to Shannon's theorem (*Wu et al., 2021*; *Xu et al., 2019*). We used $w_j$ to represent the capacity of each communication channel in edge server $e_j$.

For satisfying the user requirements, $V$ CSs ($v_k, 1 \leq k \leq V$) need to be rented from the cloud when edge resources are not enough. For CS $v_k$, the computing capacity is $g_k^v$. The storage capacity of the cloud is assumed to be infinity, so the cloud can provide all services that users request. In general, a CS is equipped with one network interface card (NIC) for the network connection. We use $w^C$ to represent the network capacity of each CS for the data transmission of tasks that are offloaded to the cloud.

There are $T$ tasks requested by all of users in the system, represented by $t_i, 1 \leq i \leq T$. For task $t_i$, it has $a_i$ input data amount should be processed by its requested service, and requires $f_i$ computing resources for its completion. Then, if task $t_i$ is offloaded to $e_j$ at $m^{th}$ channel, the data transmission latency is $a_i/w_j$, and the computing latency is $f_i/g_j$. The deadline of $t_i$ is $d_i$, which means $t_i$ must be finished before $d_i$. In this article, we focus on the hard deadline requirements of tasks, and leave soft deadline constraints as our future work. For each task, it can be only offloaded to the ES that has connection with its device. We use binary constants $o_{i,j}, 1 \leq i \leq T, 1 \leq j \leq E$, to represent these connectivities, where $o_{i,j} = 1$ if $t_i$ can be offloaded to $e_j$, and otherwise, $o_{i,j} = 0$.

In total, the system provides $S$ kinds of services ($s_l, 1 \leq l \leq S$) for its users. For $s_l$, it requires $h_l$ storage space. We use binary constants $r_{i,l}$ ($1 \leq i \leq T, 1 \leq l \leq S$) to identify the service requested by each task. When task $t_i$ requests service $s_l$, $r_{i,l} = 1$.

For the formulation of the joint service caching and task offloading problem, we define binary variables $x_{i,j,m}$ ($1 \leq i \leq T, 1 \leq j \leq E, 1 \leq m \leq N_j$) to represent offloading decisions and communication channel allocations in ESs, as shown in Eq. (1), and use $y_{i,k}$ ($1 \leq i \leq T, 1 \leq k \leq V$) to represent the offloading decisions in the cloud as shown in Eq. (2). For each task, it can be offloaded to only one ES/CS, and accesses only one channel when it is offloaded to an ES, thus, in Eqs. (3) hold.

$$x_{i,j,m} = \begin{cases} 1, & \text{if } t_i \text{ is offloaded to } e_j \\ & \text{and the } m^{th} \text{ channel is allocated for the task's data transmission} \,. \\ 0, & \text{else} \end{cases} \quad (1)$$

$$y_{i,k} = \begin{cases} 1, & \text{if } t_i \text{ is offloaded to } v_k \\ 0, & \text{else} \end{cases} \,. \quad (2)$$

$$\sum_{j=1}^{E}\sum_{m=1}^{N} x_{i,j,m} + \sum_{k=1}^{V} y_{i,k} \leq 1, i = 1, ..., T. \tag{3}$$

To make following formulations more concise, we define binary variables $x_{i,j}^{e}$ ($1 \leq i \leq T, 1 \leq j \leq E$) as the offloading decisions of tasks to ESs, where $x_{i,j}^{e} = 1$ if $t_i$ is offloaded to $e_j$, and otherwise $x_{i,j}^{e} = 0$. Then $x_{i,j}^{e} = 1$ if and only if there is a channel allocated to $t_i$ on $e_j$, i.e., $\sum_{m=1}^{N_j} x_{i,j,m} = 1$. Thus, Eq. (4) are established. A task can be offloaded to an ES only if the ES has connection with its device, which can be formulated as Eq. (5).

$$x_{i,j}^{e} = \sum_{m=1}^{N_j} x_{i,j,m}, i = 1, ..., T, j = 1, ..., E. \tag{4}$$

$$x_{i,j}^{e} \leq o_{i,j}, i = 1, ..., T, j = 1, ..., E. \tag{5}$$

## Task processing model in edge

When a task is offloaded to an ES for its processing, its requested service has been cached on the ES. In such situation, the task transmits its input data to the ES on the channel allocated to the task (the $m^{th}$ channel such that $x_{i,j,m} = 1$) for its computing. After the input data delivery is completed, the service will process these input data by computing resources of the ES.

For $t_i$ offloaded to $e_j$ for processing, its finish time of the input data transmission is the start time of the transmission plus the transmission time. This can be formulated as Eq. (6), where $q_i$ and $p_i$ respectively represent the finish time and start time of the input data transmission for $t_i$ offloaded to an ES.

$$\sum_{j=1}^{E}\sum_{m=1}^{N} (x_{i,j,m} \cdot q_i) = \sum_{j=1}^{E}\sum_{m=1}^{N} (x_{i,j,m} \cdot (p_i + a_i/w_j)), i = 1, ..., T. \tag{6}$$

For each channel on an ES, it is usually allocated to multiple tasks for their data transmission. To avoid the interference, we exploit the sequential data transmission model for these tasks on a channel. For any two tasks, say $t_{i1}$ and $t_{i2}$, the input data transmission of one task can be only started after finishing another's, when they both use one channel on one ES. If the transmission of $t_{i1}$ is started before that of $t_{i2}$, we have $p_{i1} \leq q_{i1} \leq p_{i2} \leq q_{i2}$. Otherwise, $p_{i2} \leq q_{i2} \leq p_{i1} \leq q_{i1}$. Thus, we have constraints (7) that must be satisfied.

$$\sum_{j=1}^{E}\sum_{m=1}^{N} (x_{i1,j,m} \cdot x_{i2,j,m} \cdot (q_{i1} - p_{i2}) \cdot (q_{i2} - p_{i1})) \leq 0, i1, i2 = 1, ..., T. \tag{7}$$

Similar to the data transmission constraints, the finish computing time of a task is its start computing time plus its computing latency. Thus, Eq. (8), where $cst_i$ and $z_i$ are respectively the start time and the finish time of computing task $t_i$ offloaded to an ES.

As the computing of a task can be started only after the finish of its input data transmission, constraints (9) must be met.

$$\sum_{j=1}^{E} (x_{i,j}^e \cdot z_i) = \sum_{j=1}^{E} (x_{i,j}^e \cdot (c_i + f_i/g_j)), i = 1, ..., T. \tag{8}$$

$$\sum_{j=1}^{E} \sum_{m=1}^{N} (x_{i,j,m} \cdot z_i) \leq \sum_{j=1}^{E} (x_{i,j}^e \cdot c_i), i = 1, ..., T. \tag{9}$$

Also, to avoid the interference, we assume tasks are computed sequentially, and thus, similar to constraints (7), we have constraints (10).

$$\sum_{j=1}^{E} (x_{i1,j}^e \cdot x_{i2,j}^e \cdot (z_{i1} - c_{i2}) \cdot (z_{i2} - c_{i1})) \leq 0, i1, i2 = 1, ..., T. \tag{10}$$

When a task offloaded to an ES, its requested service must be cached in the ES. The storage capacity of an ES limits the number of services. This can be formulated into Eq. (11).

$$\sum_{i=1}^{T} (x_{i,j}^e \cdot r_{i,l} \cdot h_l) \leq b_j, j = 1, ..., E. \tag{11}$$

## Task processing model in cloud

When a task is offloaded to the cloud, there are mainly two ways for using CSs. One use is that each cloud server is monopolized by a task. Another method of use is that a CS can be multiplexed by multiple tasks. The first method of use has no data transmission wait time for tasks. However, this will increase the cost of using CSs, because the CS is charged on a per-unit-time basis. For example, when a CS is used only 1.1 h, it is charged 2 h. Thus, we exploit the second way for multiplexing CSs to improve the resource and cost efficiencies. In such way, the formulation of the task processing in the cloud is similar to that in edges. When a task is offloaded to a CS, its finish time of the input data transmission and the computing can be calculated by Eqs. (12) and (13), respectively. Where $q_i^C$ ($p_i^C$) and $z_i^C$ ($c_i^C$) are respectively the finish time (the start time) of the input data transmission and the computing of $t_i$ offloaded to a CS, respectively.

$$\sum_{k=1}^{V} (y_{i,k} \cdot q_i^C) = \sum_{k=1}^{V} (y_{i,k} \cdot (p_i^C + a_i/w^C)), i = 1, ..., T. \tag{12}$$

$$\sum_{k=1}^{V} (y_{i,k} \cdot z_i^C) = \sum_{j=1}^{E} (y_{i,k} \cdot (c_i^C + f_i/g_k^v)), i = 1, ..., T. \tag{13}$$

To avoid interferences, when multiple tasks are offloaded to one CS, the CS processes the data transmission and the computing sequentially. Thus, constraints (14) and (15) must be satisfied, similar to constraints (7) and (10).

$$\sum_{k=1}^{V} (y_{i1,k} \cdot y_{i2,k} \cdot (q_{i1}^C - p_{i2}^C) \cdot (q_{i2}^C - p_{i1}^C)) \leq 0, i1, i2 = 1, ..., T. \tag{14}$$

$$\sum_{k=1}^{V} (y_{i1,k} \cdot y_{i2,k} \cdot (z_{i1}^C - c_{i2}^C) \cdot (z_{i2}^C - c_{i1}^C)) \leq 0, i1, i2 = 1, ..., T. \tag{15}$$

## Joint service caching and task offloading problem model

Based on the edge-cloud system and the task processing models, we can formulate the joint service caching and task offloading problem as the following mix-integer non-linear programming (MINLP).

**Maximizing** $N_{fin} + U$. $\tag{16}$

Subject to,

$$Eq.(4) - (15), \tag{17}$$

$$N_{fin} = \sum_{i=1}^{T} (\sum_{j=1}^{E} x_{i,j}^e + \sum_{k=1}^{V} y_{i,k}), \tag{18}$$

$$U = \frac{\sum_{i=1}^{T} ((\sum_{j=1}^{E} x_{i,j}^e + \sum_{k=1}^{V} y_{i,k}) * f_i)}{\sum_{j=1}^{E} (\max_{i=1}^{T} (x_{i,j}^e * z_i) * g_j) + \sum_{k=1}^{V} \left( \left\lceil \frac{\max_{i=1}^{T} (y_{i,k} * z_i^C)}{3600} \right\rceil * 3600 * g_k^v \right)}, \tag{19}$$

$$z_i \leq d_i, i = 1, ..., T, \tag{20}$$

$$z_i^C \leq d_i, i = 1, ..., T, \tag{21}$$

$$x_{i,j,m} \in \{0, 1\}, i = 1, ..., T, j = 1, ..., E, m = 1, ..., N_j, \tag{22}$$

$$y_{i,k} \in \{0, 1\}, i = 1, ..., T, k = 1, ..., V. \tag{23}$$

There are two optimization objectives in our problem formulation, as shown in Eq. (16), which are maximizing the number of tasks whose requirements are satisfied ($N_{fin}$) and the computing resource utilization ($U$). $N_{fin}$ is the accumulated number of tasks finished in the edge-cloud computing system, which is one of popular metrics quantifying the user satisfaction and can be calculated by Eq. (18). $U$ is the ratio between the amount of computing resources used by tasks and that of occupied resources. The amount of computing resources used by tasks is the accumulated amount of computing resources required by finished tasks, which is $\sum_{i=1}^{T} \left( \left( \sum_{j=1}^{E} x_{i,j}^e + \sum_{k=1}^{V} y_{i,k} \right) * f_i \right)$. For each ES/CS, its occupied time is the latest finish time of tasks offloaded to it. Thus, for an ES $e_j$, the occupied time is $\max_{i=1}^{T} (x_{i,j}^e * z_i)$, and its occupied computing resource amount is $\max_{i=1}^{T} (x_{i,j}^e * z_i) * g_j$. For a CS, its occupied time is the ceiling time of the latest finish time of tasks offloaded to it, because CSs are charged on a per-unit-time basis. In this article, we assume CSs are charged on hour, as done by Amazon ES2, AliCloud, *etc.* Thus, for CS $v_k$,

---

**Algorithm 1** Multi-stage heuristic joint service caching and task offloading method (MSHCO).

**Input:** The information of tasks and resources in the edge-cloud computing system.

**Output:** A joint service caching and task offloading strategy.

1: Pre-offloading tasks whose requirements can be satisfied by cloud resources to the cloud, using Algorithm 2;

2: For tasks that are not pre-offloaded to the cloud, caching their requested services on and offloading them to edges by Algorithm 3;

3: Re-offloading tasks that have pre-offloaded to the cloud from the cloud to edges by Algorithm 4;

4: **return** The joint service caching and task offloading strategy;

---

the occupied time is $\left\lceil \dfrac{\max_{i=1}^{T}(y_{i,k} * z_i^C)}{3600} \right\rceil * 3600$ and its occupied computing resource amount is $\left\lceil \dfrac{\max_{i=1}^{T}(y_{i,k} * z_i^C)}{3600} \right\rceil * 3600 * g_k^y$. And therefore, the overall computing resource utilization can be calculated by Eq. (19). Noticing that $U$ is no more than 1, the user satisfaction maximization is the major optimization objective, and the computing resource utilization maximization is the minor one. Constraints (17) mainly include the calculation of tasks' finish time and the deadline requirements. Constraints (20) and (21) restrict the deadline of each task. Constraints (22) and (23) represent that each task can be only offloaded to only one ES/CS for its processing.

## Hardness of the optimization problem

For an instance of our optimization problem, there are limitless commutation channels and storage capacity for each ES. The problem instance is identical to the task scheduling on multiple heterogeneous cores by seeing each ES/CS as a computing core, which has been proofed as NP-hard problem (*Gary & Johnson, 1979*). Thus, our optimization problem is NP-hard. The optimization problem is MINLP due to the non-liner constraints (*e.g.*, Eq. (19)). Existing tools, *e.g.*, lp_solve (*Berkelaar, Eikland & Notebaert, 2021*), can be used for solving the problem based on simple branch and bound. But these tools need exponential time, and thus is not applicable to large-scale system. Therefore, we propose a heuristic method in the following section to achieve an approximate solution.

## MULTI-STAGE HEURISTIC METHOD

In this section, we propose a polynomial time heuristic method with three stages, outlined in Algorithm 1, for solving the joint service caching and task offloading problem presented in the previous section. We abbreviate the proposed method to MSHCO. In the first stage, MSHCO employs the abundance of cloud resources, by pre-offloading all tasks to the cloud considering deadline constraints. For tasks that their deadline constraints can not ensured by the cloud, MSHCO caches their requested services on edges and offloads them to edges. This is the second stage of MSHCO, which exploits limited edge resources for satisfying requirements of latency-sensitive tasks. At the last stage, to take full advantage of edge resources providing low latency network, MSHCO re-offloads pre-offloaded tasks from the cloud to edges, to improve the overall performance of task processing. In the

**Table 2 The symbols used in algorithm illustrations**

| Symbol | Description |
|---|---|
| $\mathbb{T}$ | The set of tasks $\{t_i | i = 1, \ldots, T\}$. Each task, say $t$, has following attributes: the required computing resource size ($t.f$), the input data amount ($t.a$), the deadline ($t.d$), the requested service ($t.s$), the set of edge servers having network connections with its device ($t.\mathbb{E}$). |
| $\mathbb{S}$ | The set of services $\{s_l | l = 1, \ldots, S\}$. Each service has one attribute, which is its required storage space, $s.b$. |
| $\mathbb{E}$ | The set of ESs $\{e_j | j = 1, \ldots, E\}$. Each ES has following attributes: the computing capacity ($e.g$), the number of communication channels ($e.N$), the communication channel capacity ($e.w$), the set of tasks offloaded to the ES with each channel ($e.\mathbb{T}_m$, $m = 1, \ldots, e.N$), the set of services cached on the ES ($e.\mathbb{S}$). |
| $\mathbb{VT}$ | The set of CS types, $\{vt_o | o = 1, \ldots, VT\}$, provided by the cloud. Each type has following configuration parameter: the computing capacity ($vt.g$), the network transmission capacity ($vt.w$), the price per unit time ($vt.p$). |
| $\mathbb{V}$ | The set of rented CSs $\{v_k | k = 1, \ldots, V\}$. Each CS has following attributes: the configuration type ($v.vt$), the set of tasks offloaded to the CS ($v.\mathbb{T}$). |

**Algorithm 2 Stage 1: pre-offloading tasks to the cloud.**

**Input:** tasks to be offloaded, $\mathbb{T}$; **CS** types provided by the cloud, $\mathbb{VT}$;

**Output:** rented **CSs** for satisfying requirements of pre-offloaded tasks, and a task pre-offloading strategy in the cloud, $\mathbb{V}$; tasks whose requirements cannot satisfied by the cloud, $\mathbb{T}$;

1:   **for each** $t \in \mathbb{T}$ **do**

2:       **for each** $v \in \mathbb{V}$ **do**

3:           calculating the finish time $ft$ of $t$ if it is offloaded to $v$ based on Eqs. (12)–(15);

4:           **if** $ft \leq t.d$ **then**

5:               $v.\mathbb{T} \leftarrow v.\mathbb{T} \cup \{t\}$; //pre-offloading $t$ to the rented **CS**

6:               $\mathbb{T} \leftarrow \mathbb{T} - \{t\}$; //removing $t$ from un-offloaded task set

7:               goto line 1;

8:       **for** each $vt \in \mathbb{VT}$ **do**

9:           calculating the finish time $ft$ of $t$ if it is offloaded to a CS with type $vt$ based on Eqs. (12)–(15);

10:          **if** $ft \leq t.d$ **then**

11:              pre-rent a **CS** $v$, where $v.vt \leftarrow vt$; //renting a **CS** with type $vt$

12:              same to lines 5 – 7;

13:  **return** $\mathbb{V}$, $\mathbb{T}$;

following, we illustrate the details of MSHCO. Table 2 gives some symbols used in following illustrations.

As shown in Algorithm 2, in the first stage, MSHCO makes a decision for pre-offloading tasks to the cloud. For each task to be offloaded, MSHCO examines whether the task can be finished before its deadline by a CS that has been rented from the cloud (lines 3–4). If there is one such CS, MSHCO pre-offloads the task to the CS (lines 5–6), and repeats these steps (of lines 3–6) for the next task (line 7). Otherwise, MSHCO tries to rent a new CS for the task (lines 8–12). MSHCO finds a CS type with the resource configuration that satisfies the task's requirements, and rents a new CS with the type (line 11). Then, MSHCO pre-offloads the task to the new CS (line 12). If no rented CS or CS type can be found for finishing the task within its deadline, the task can not be offloaded to the cloud,

---

**Algorithm 3** Stage 2: offloading tasks to edges.

**Input:** tasks that are not pre-offloaded to the cloud, $\mathbb{T}$; services, $\mathbb{S}$; edge servers, $\mathbb{E}$;

**Output:** A joint service caching and task offloading strategy in **ESs**, $\{e.\mathbb{T}_m, e.\mathbb{S} \mid e \in \mathbb{E}, m = 1, \ldots, e.N\}$.

1:    **for** each $t \in \mathbb{T}$ **do**
2:      **for** each $e \in t.\mathbb{E}$ **do**
3:        **if** $t.s \in e.\mathbb{S}$ **then**
4:          **for** each channel of $e$, $m = 1$ to $e.N$ **do**
5:            calculating the finish time $ft$ of $t$ if it is offloaded to $e$ at $m^{th}$ channel based on Eqs. (6)–(10);
6:            **if** $ft \leq t.d$ **then**
7:              $e.\mathbb{T}_m \leftarrow e.\mathbb{T}_m \cup \{t\}$; //offloading $t$ to the ES having cached the requested service
8:              goto line 1;
9:        **else if** $t.s.b + \sum_{s \in e.\mathbb{S}} s.b \leq e.b$ **then**
10:         **for** each channel of $e$, $m = 1$ to $e.N$ **do**
11:           calculating the finish time $ft$ of $t$, same to line 5;
12:           **if** $ft \leq t.d$ **then**
13:             $e.\mathbb{S} \leftarrow e.\mathbb{S} \cup \{t.s\}$; //caching the service on the ES
14:             same to lines 7 – 8;
15:   **return** $\mathbb{E}$;

---

and MSHCO examines the next task with previous procedures. After executing Algorithm 2, we achieve the CSs needed to be rented from the cloud, and the tasks pre-offloaded to each CS, stored in $\mathbb{V}$ (line 13).

The second stage of MSHCO is illustrated in Algorithm 3. For each task that is not pre-offloaded to the cloud (line 1), MSHCO finds an ES that can receive the request and has enough resources for processing the task. There are two situations when offloading a task to an ES, the requested service has or not cached on the ES. When the requested service has cached on the ES, it is only needed to examine whether the task can be finished within its deadline using one of communication channels (lines 3–8). If so, the task is offloaded to the ES (lines 6–8), and otherwise, the task's requirements cannot be satisfied by edge resources. When the requested service has not cached for the current examined task, MSHCO sees whether the ES has enough storage space for caching the requested service (line 9). If the ES has enough storage space and the task's deadline constraint can be met by the ES (line 12), the requested service will be cached on the ES (line 13), and the task is offloaded to the ES (line 14). After execution, Algorithm 3 provides a task offloading solution and a service caching solution in edges.

Benefiting from Algorithms 2 and 3, we have a strategy for jointly task offloading and service caching on the edge-cloud computing system. Because we first exploit cloud resources for task offloading, some tasks pre-offloaded to the cloud can be processed by edge resources. Usually edge resources provide a better network performance than cloud resources. Thus, MSHCO re-offloads these tasks from the cloud to edges in the last stage to

---

**Algorithm 4** Stage 3: re-offloading tasks from the cloud to edges.

**Input:** the service caching and task (pre-)offloading strategy in the cloud and edges, $\mathbb{V}$ and $\mathbb{E}$;

**Output:** An improved joint service caching and task offloading strategy, updated $\mathbb{V}$ and $\mathbb{E}$.

1:   **for** each $v \in \mathbb{V}$ **do**

2:     **for** each $t \in v.\mathbb{T}$ **do**

3:       **for** each $e \in t.\mathbb{E}$ **do**

4:         **if** $t.s \in e.\mathbb{S}$ **then**

5:           **for** each channel of $e$, $m = 1$ to $e.N$ **do**

6:             calculating the finish time $ft$ of $t$ if it is offloaded to $e$ at $m^{th}$ channel based on Eqs. (6)–(10);

7:             **if** $ft \leq t.d$ **then**

8:               $e.\mathbb{T}_m \leftarrow e.\mathbb{T}_m \cup \{t\}$; //offloading $t$ to the **ES** having cached the requested service

9:               $v.\mathbb{T} \leftarrow v.\mathbb{T} - \{t\}$;

10:              goto line 2;

11:         **else if** $t.s.b + \sum_{s \in e.\mathbb{S}} s.b \leq e.b$ **then**

12:           **for** each channel of $e$, $m = 1$ to $e.N$ **do**

13:             calculating the finish time $ft$ of $t$, same to line 6;

14:             **if** $ft \leq t.d$ **then**

15:               $e.\mathbb{S} \leftarrow e.\mathbb{S} \cup \{t.s\}$; //caching the service on the **ES**

16:               same to lines 8 – 10;

17:   **return** $\mathbb{E}$, $\mathbb{V}$;

improve the overall performance of task processing, as shown in Algorithm 4. In this stage, MSHCO examines whether each task having been pre-offloaded to the cloud can be processed by ESs (lines 1–2). The examining procedures are identical to Algorithm 2, except that when a task can be offloaded to an ES, the task will be removed from the CS that the task is pre-offloaded to (lines 9 and 16).

The main advantage of our method is that all tasks whose requirements can be satisfied by the cloud are pre-offloaded to the cloud. This can result in more edge resources for finishing more delay-sensitive tasks, compared with other methods that offload tasks to the cloud only when edge resources are exhausted. Here, we give an example to illustrate the advantage of our method. Considering a simple edge-cloud environment, there are four tasks ($t_1$, $t_2$, $t_3$, and $t_4$), 1 ES, and 1 CS type. The input data amount and the required computing size of $t_1$ and $t_2$ are 1 MB and 20 MHz, respectively. The input data amount and the required computing size of $t_3$ and $t_4$ are 2 MB and 40 MHz, respectively. The deadline of each task is set as 110 ms. All of services requested by these tasks are all deployed on the ES. The communication and computing capacities of the ES are 100 MB/s and 2 GHz, respectively. The network and computing capacities configured by the CS type are 10 MB/s and 2 GHz, respectively. The time consumed by $t_1$ or $t_2$ ($t_3$ or $t_4$) is 20 ms (40 ms) when it is offloaded to the ES, and 110 ms (220 ms) when offloaded to the cloud. In this case, if we offload tasks to the ES first, $t_1$, $t_2$, and $t_3$ are offloaded to ES with finish

time of 80 ms, and $t_4$ is rejected. But by MSHCO, $t_1$ and $t_2$ are pre-offloaded to two CS rented from the cloud, at the first stage. $t_3$ and $t_4$ are offloaded to the ES at the second stage, and thus all tasks can be finished within their respective deadlines. We can see that MSHCO can satisfy requirements of more tasks. The third stage only changes the locations where some tasks are offloaded, and has no effect on which tasks are offloaded. Therefore, the user satisfaction is not changed by the third stage.

### Algorithmic complexity analysis

In the first stage, MSHCO visits each rented CSs and CS types for each task. Thus, this stage has $O(T * (V + VT))$ time complexity at worst. In real word, the number of CS types provided by the cloud is very limited, and can be considered as a small constant. Thus, the time complexity of the first stage is $O(T * V)$. In the second stage, for a task, MSHCO searches all channels of ESs having connection with its device, and examines whether its requested service has been cached. In general, there are only a few edges connecting its device for each task. Therefore, the time complexity of the second stage is $O(T * S)$. Similar to the second stage, the third stage has a time complexity of $O(T * S)$, too. Thus, in overall, the time complexity of MSHCO is $(T * (V + S))$, which is increased linearly with the numbers of tasks, rented CSs, and services in the edge-cloud computing system.

## PERFORMANCE EVALUATION

In this section, we evaluate the performance of MSHCO by conducting extensive simulated experiments. The simulated system is established referring to recent related works (*Dai et al., 2019*; *Xia et al., 2022*; *Zhang et al., 2021a*) and CS types provided by Amazon EC2 (*Amazon.com, 2022*). Specifically, in the simulated system, there are 10 ESs and 1 CS type. Each ES has 20 GHz computing capacity, 10 communication channels, and 300 GB storage capacity. The communication capacity of each communication channel is set as 60 Mbps. The CS type has the configurations of 5 GHz computing capacity and 15 Mbps network transfer capacity, with the price of \$ 0.1 per h. There are 1,000 tasks in the simulated system. Each task randomly requests one of 100 services. The input data amount and required computing resource size of a task is randomly set in the ranges of [1.5, 6] MB and [0.5, 1.2] GHz. The deadlines of tasks are randomly set in range [1, 5] seconds. For each task, its device has a connection with randomly selected one ES. The storage space required by a service is randomly set in the range of [40, 80] GB.

To show the advantage of our method, we select the following classical and recently published works for performance comparison.

- **First fit (FF)** is one of the classical and most widely used methods in various computing systems. FF iteratively offloads each task to the ES/CS that can satisfy all requirements of the task.
- **First fit decreasing (FFD)** is identical to FF except that FFD makes offloading decision for the task with maximal computing resource size in each iteration.
- **Earliest deadline first (EDF)** is one of deadline-aware method. EDF iteratively makes offloading decision for the task with earliest deadline.

**Table 3 The finished task number achieved by various methods in several system instances.**

| Time | FF | FFD | EDF | PSC | CEDC-A | MSHCO |
|---|---|---|---|---|---|---|
| 1 | 192 (1.000) | 192 (1.000) | 191 (0.995) | 173 (0.901) | 119 (0.620) | 206 (1.073) |
| 2 | 174 (1.000) | 174 (1.000) | 174 (1.000) | 154 (0.885) | 85 (0.489) | 186 (1.069) |
| 3 | 186 (1.000) | 186 (1.000) | 191 (1.027) | 182 (0.978) | 108 (0.581) | 210 (1.129) |
| 4 | 178 (1.000) | 178 (1.000) | 183 (1.028) | 164 (0.921) | 110 (0.618) | 208 (1.169) |
| 5 | 176 (1.000) | 176 (1.000) | 179 (1.017) | 155 (0.881) | 98 (0.557) | 190 (1.080) |
| 6 | 193 (1.000) | 193 (1.000) | 194 (1.005) | 172 (0.891) | 111 (0.575) | 225 (1.166) |
| 7 | 185 (1.000) | 185 (1.000) | 183 (0.990) | 176 (0.951) | 106 (0.573) | 202 (1.092) |
| 8 | 198 (1.000) | 198 (1.000) | 201 (1.015) | 178 (0.899) | 115 (0.581) | 231 (1.167) |
| 9 | 172 (1.000) | 172 (1.000) | 175 (1.017) | 155 (0.901) | 96 (0.558) | 187 (1.087) |
| 10 | 185 (1.000) | 185 (1.000) | 184 (0.995) | 174 (0.941) | 107 (0.578) | 199 (1.076) |
| ... | | | | | | |
| 100 | 175 (1.000) | 175 (1.000) | 174 (0.994) | 168 (0.960) | 98 (0.560) | 193 (1.103) |

Note:
The data format: the number of finished tasks (the value scaled by that of FF).

- **Popularity-based service caching (PSC)** is the basis idea exploited by *Wei et al. (2021)*. PSC caches the service with maximal requests on ESs. For offloading decisions, it uses FF strategy.
- **Approximation algorithm for constrained edge data caching (CEDC-A)** (*Xia et al., 2021b*) caches a service on an ES, such that the caching solution provides maximum benefit. The benefit is quantified based on the number of network hops. For our comparison, we set the benefit value as the accumulated slack time for each caching solution, where the slack time is the different between the deadline and the finish time for a task.

We evaluate the performance of these methods in the following aspects. For following each metric, a higher value is better.

- **User satisfaction** strongly affects the income and the reputation of service providers. We use three metrics for its quantification, the number of finished tasks[2], the computing resource size of finished tasks, and the processed input data amount of finished tasks.
- **Resource efficiency** is closely related to the cost of service provision. We use the computing resource utilization for quantifying the resource efficiency.
- **Processing efficiency** is representing the workload processing rate, which can determine the user satisfaction and the resource efficiency at a large extent. We use two metrics for the quantification, the computing rate and the data processing rate, which are the size of computing finished per unit time and the amount of input data processed per unit time, respectively.

For conducting the experiment, we randomly generate 100 edge-cloud system instances. For each instance, we achieve a set of performance values for each method, and scale each performance value of a method by that of FF for each metric. For example, Table 3 shows a

[2] A finished task means the task's requirements are met in the computing system.

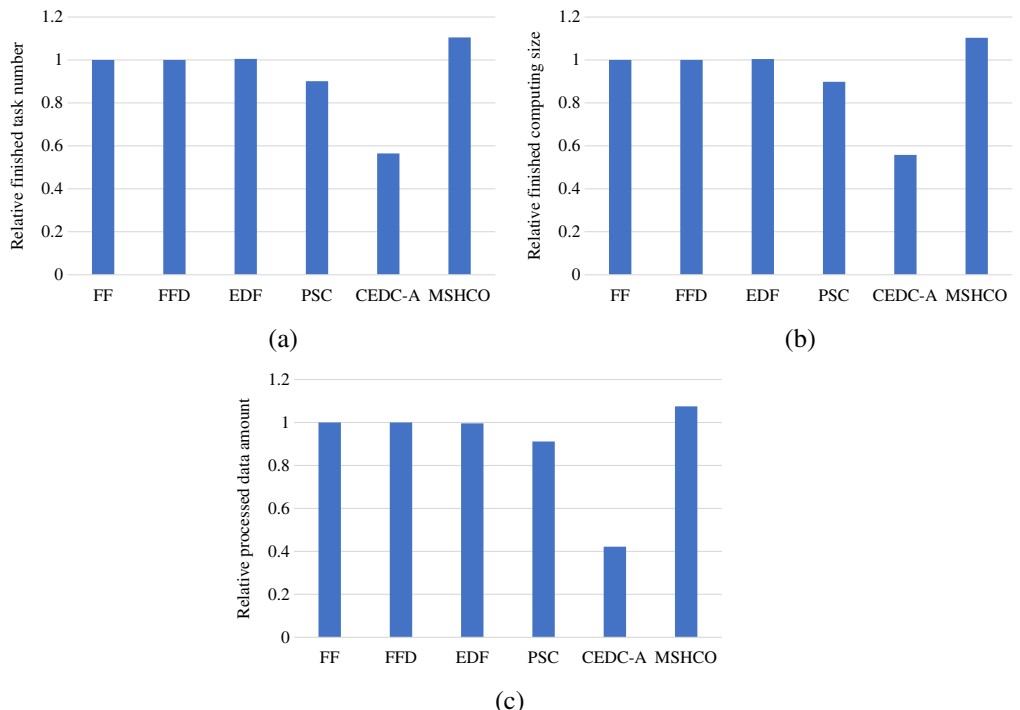

**Figure 2 The overall user satisfaction achieved by various methods.** (A) Relative number of finished tasks. (B) Relative computing size of finished tasks. (C) Relative amount of processed input data.

part of the experiment data in the metric of finished task number. In the following, we report the average relative value for each metric.

In addition, we compare the solution solved by our method with the optimal solution in simulated mini edge-cloud systems, which is illustrated in the end of this section.

## Comparison in user satisfaction

Figure 2 shows the overall user satisfaction achieved by various methods in various performance metrics. As shown in the figure, we can see that MSHCO has 9.94–95.6%, 9.85–97.8%, and 7.97–155% better overall user satisfaction, compared with other methods, respectively in the number, the computing size, and the input data amount of finished tasks. This result verifies that our proposed method has good effect on optimizing the user satisfaction. The main reason is that our method prioritizes the use of abundant cloud resources for processing tasks. This can avoid the situation that tasks whose requirements can be satisfied by the cloud occupy limited edge resources at first, and thus leaves more edge resources for latency-sensitive tasks. While other methods employ cloud resources only after exhausting all edge resources resources.

Thus, in most of time, the improvement of our method in edges is greater than that in the cloud in optimizing user satisfaction. For example, compared with PSC, MSHCO has 14.8%, 15.1%, and 10.3% greater values in three user satisfaction metrics in edges, respectively, as shown in Fig. 3. But MSHCO has 32.3%, 32.4%, and 35.0% greater values in these three metrics in the cloud, respectively, as shown in Fig. 4. A similar result can be

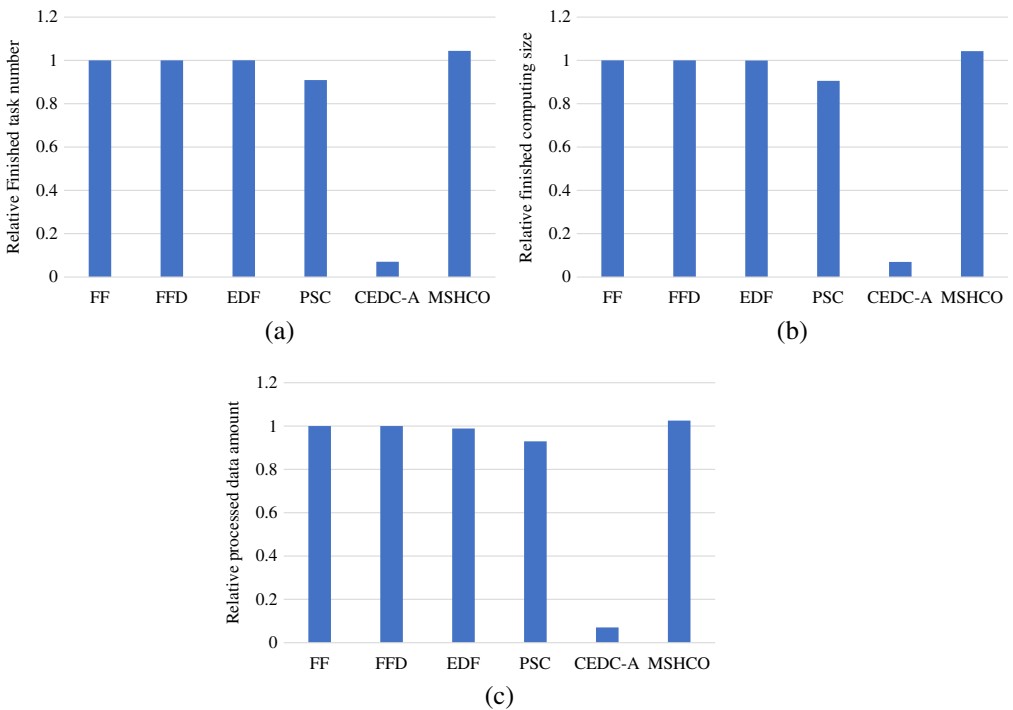

**Figure 3 The user satisfaction achieved by various methods in edges.** (A) Relative number of finished tasks. (B) Relative computing size of finished tasks. (C) Relative amount of processed input data.

seen from Fig. 3 and Fig. 4, by comparing MSHCO with FF, FFD, or EDF. While, there is an opposite behaviour for MSHCO *vs.* CEDC-A. MSHCO has about 1,370% greater value in edges but only about 0.974% greater value in the cloud, in each user satisfaction metric, as shown in Figs. 3 and 4. This is because CEDC-A offloads too less tasks to edges, compared with other methods. This can lead to more remaining tasks with requirements that can be satisfied by the cloud after conducting task offloading on edges. Meantime, MSHCO re-offloads some tasks from the cloud to edges for improving the overall performance of task processing, which results in more tasks offloaded to edge but less tasks offloaded to the cloud.

Tables 4–6 show the statistical data of user satisfactions achieved by various methods in the three metrics, respectively. From these tables, we can see that MSHCO achieves the greatest minimal value for every metric. This means that MSHCO achieves the best user satisfaction any time. This further confirms the superior performance of our method in optimizing user satisfaction. The main benefit of MSHCO is the idea of making offloading decisions by three stages, which can be applied to any task scheduling method to improve its performance for edge-cloud computing. For a task scheduling method, the use of three-stage idea ensures a performance no worse than the scheduling method used alone. This is mainly because the first stage guarantees all tasks can be finished if their requirements can be satisfied by the cloud, without any edge resources. This enables the limited edge resources to finish tasks that cannot be satisfied by the cloud at first, in the second stage. This can lead to more finished tasks that can only be finished by edge resources.

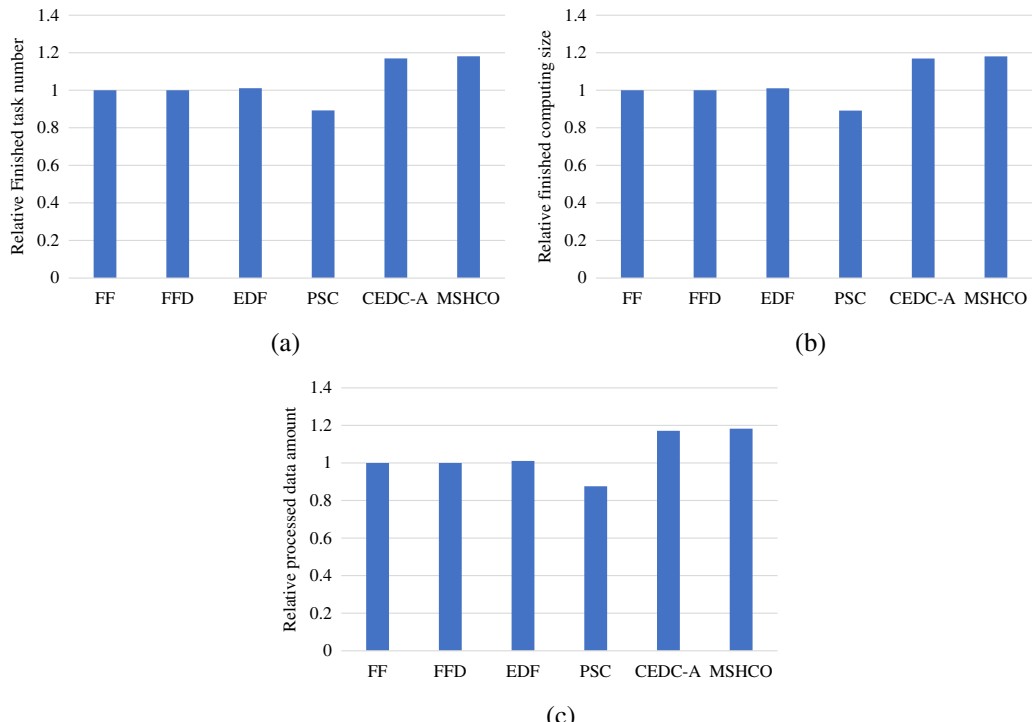

**Figure 4 The overall user satisfaction achieved by various methods in the cloud.** (A) Relative number of finished tasks. (B) Relative computing size of finished tasks. (C) Relative amount of processed input data.

**Table 4 The statistical data of the relative finished task number.**

|  | FF | FFD | EDF | PSC | CEDC_A | MSHCO |
|---|---|---|---|---|---|---|
| MAXIMUM | 1 | 1 | 1.0514 | 0.9821 | 0.6344 | 1.2216 |
| AVERAGE | 1 | 1 | 1.0049 | 0.9009 | 0.5647 | 1.1048 |
| MINIMUM | 1 | 1 | 0.9677 | 0.8305 | 0.4859 | 1.0452 |

**Table 5 The statistical data of the relative computing size of finished tasks.**

|  | FF | FFD | EDF | PSC | CEDC_A | MSHCO |
|---|---|---|---|---|---|---|
| MAXIMUM | 1 | 1 | 1.0544 | 0.992 | 0.6477 | 1.2314 |
| AVERAGE | 1 | 1 | 1.0042 | 0.8983 | 0.5576 | 1.1031 |
| MINIMUM | 1 | 1 | 0.9661 | 0.8227 | 0.4771 | 1.0363 |

**Table 6 The statistical data of the relative amount of processed data.**

|  | FF | FFD | EDF | PSC | CEDC_A | MSHCO |
|---|---|---|---|---|---|---|
| MAXIMUM | 1 | 1 | 1.048 | 1.0776 | 0.5149 | 1.1628 |
| AVERAGE | 1 | 1 | 0.9957 | 0.9116 | 0.4219 | 1.0751 |
| MINIMUM | 1 | 1 | 0.9141 | 0.8084 | 0.3446 | 1.0333 |

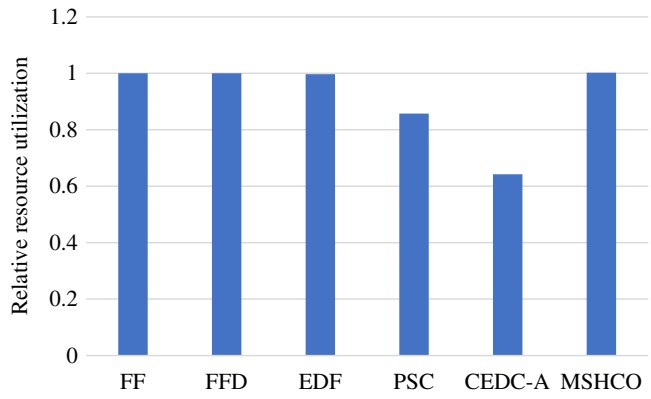

**Figure 5** **The relative overall resource utilization achieved by various methods.**

## Comparison in resource efficiency

Figure 5 shows the relative overall resource utilizations in the edge-cloud system when employing various methods for task offloading and service caching. From this figure, we can see that our method achieves 0.21%, 0.21%, 0.522%, 16.9%, and 56.1% higher resource utilizations than FF, FFD, EDF, PSC, and CEDC-A, respectively. Thus, in overall, MSHCO can achieve a better resource efficiency. More finished tasks is helpful for improving the resource utilization, because the data transfer can be performed in parallel, which is benefit for reducing the idle computing time caused by waiting the input data. Thus, as our method achieves more finished tasks than other methods, our method has a high probability in achieving better resource utilization.

In fact, the computing resource utilization is decided by the idle computing time of ESs and CSs, which is related to the relative time of the input data transfer and the task computing. Due to much poor network performance of the cloud, the resource utilization achieved in edges is usually much better than that in the cloud. In our experiment results, edge resources has more than twice utilization than cloud resources in most of time. This results in a better opportunity for improving resource utilization for the cloud than for edges. Thus, compared with FF, FFD, and EDF, MSHCO achieves negligibly lower resource utilization in edges, but about 4.8% higher in the cloud, as shown in Figs. 6 and 7. The reason why CEDC-A has much lower resource utilization in edges is mainly because it offloads too less tasks to ESs. This can give rise to that not all of communication channels of ESs are used for offloaded tasks most of time, which results in a great percentage of time that is spent on waiting input data transfer for task processing.

## Comparison in processing efficiency

Figure 8 gives the relative task processing efficiency achieved by various methods in overall. From the figure, we can see that MSHCO achieves 9.49–97.7% and 7.06–155% better processing efficiency in computing and data processing, respectively, compared with other methods. The rate of computing (or data processing) is the ratio between the accumulated computing size (or processed data amount) and the latest finish time of finished tasks. Thus, the processing efficiency is manly decided by the speedup of task parallel executions.

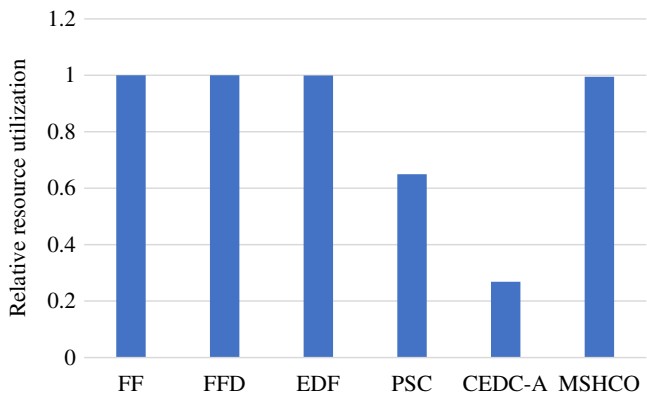

**Figure 6 The relative resource utilization achieved by various methods in edges.**

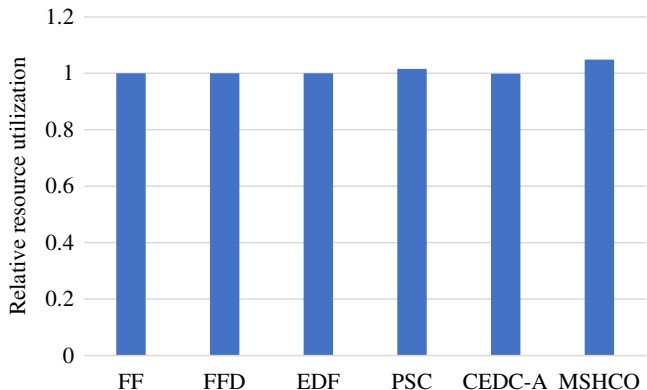

**Figure 7 The relative resource utilization achieved by various methods in the cloud.**

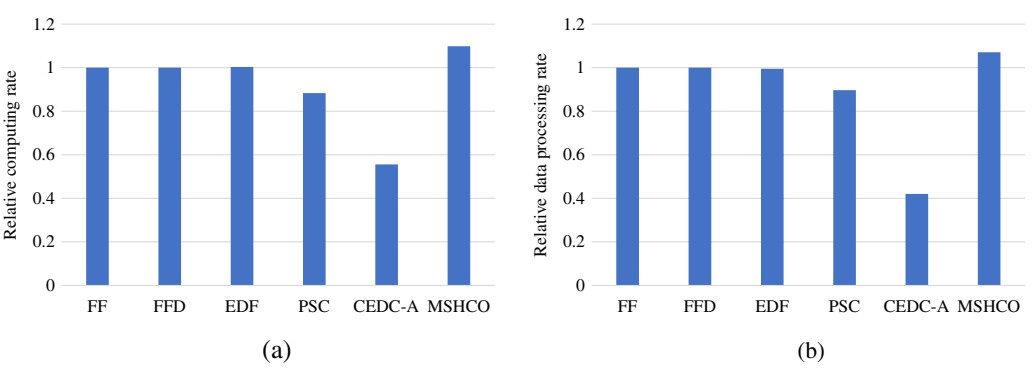

**Figure 8 The relative overall processing efficiency achieved by various methods.** (A) Relative computing rate. (B) Relative data processing rate.

In general, the resource utilization reflects the speedup to a large extent. Thus, we can deduce that MSHCO has a greater speedup from that it has a higher resource utilization in the edge-cloud system, compared with other methods. Therefore, MSHCO has a better processing efficiency in overall.

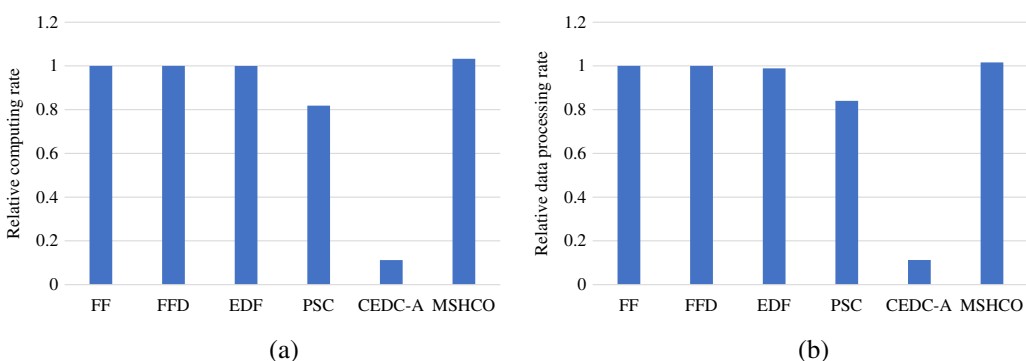

**Figure 9 The relative processing efficiency achieved by various methods in edges.** (A) Relative computing rate. (B) Relative data processing rate.

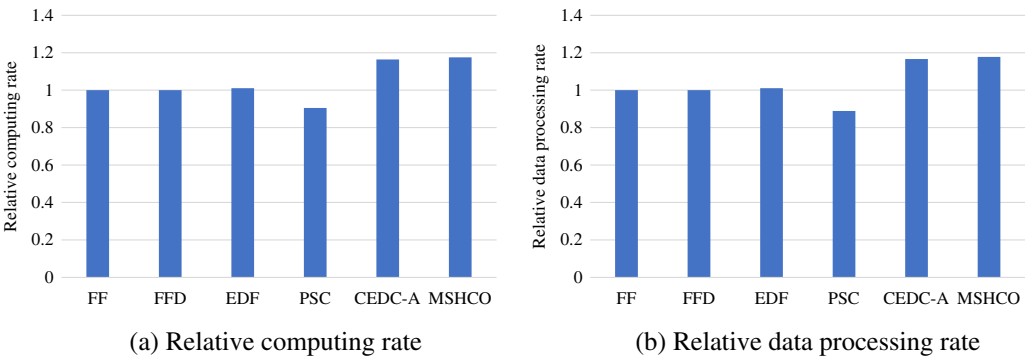

**Figure 10 The relative processing efficiency achieved by various methods in the cloud.**

In edges, An ES provides 10 communication channels for receiving input data transfers of offloaded tasks. Thus, even though the increased number of offloaded tasks helps to improve the speedup of parallel executions in edges, it aggravates the scarcity of computing resources, especially when the computing resources is bottleneck. In this case, the latest finish time of offloaded tasks can be postponed as there are more waiting time for input data transfers. While each CS receives input data by only one NIC, and thus the above issue in an edge is less serious than in the cloud. Therefore, in general, the improvement of processing efficiency in edges is less than in the cloud, as shown in Figs. 9 and 10. For example, compared with EDF, MSHCO achieves 3.32% faster computing rate in edges while 16.3% in the cloud.

## Comparison with optimal solution

To compare our method with the exhaustive method providing the optimal solution, we generate a mini-scale edge-cloud system. The mini-scale system is consisted of one ES, and one CS type, and provides two services. The ES can cache only one service in a time. We conduct 11 groups of experiments with varied number of tasks. The number of tasks in $u^{th}$ group of experiments is $8 + 2 * u$. The other parameters of the system is same to the previous experiment. Each group of experiments was repeated 11 times, and the average

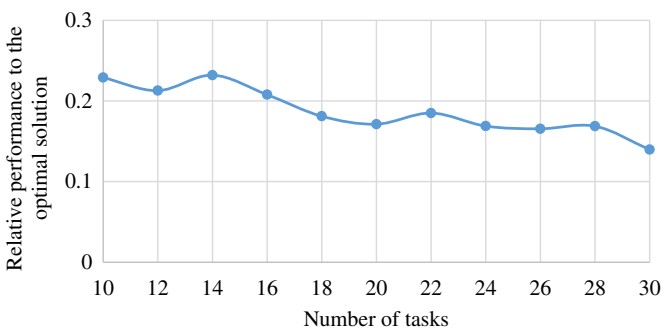

**Figure 11 The performance achieved by MSHCO, relative to the optimal solution, in the user satisfaction metric of finished task number, with varied numbers of request tasks.**

value of the relative performance of our method to the exhaustive method in finished task number was reported. The result is shown in Fig. 11.

As shown in Fig. 11, we can see that our method finishes 14.0–23.2% less tasks than the exhaustive method, and the overall trend is that the different performance between MSHCO and the exhaustive method is decreased with the increasing scale of the system. Thus, we could argue that our method can achieve a nearly optimal solution, in a large scale edge-cloud computing system.

# RELATED WORKS

The edge-cloud computing is an efficient way to address the scarcity of edge and device resources and the high latency of cloud resources. The resource efficiency of edge-cloud computing can be improved by well designed task offloading strategy. Thus, in recent years, there are several works focusing on the design of task offloading methods. For example, *Liu et al. (2021)* modelled the task offloading problem in edge-cloud computing as a Markov Decision Process (MDP), and applied deep reinforcement learning to achieve an offloading solution. *Wang et al. (2022)* presented an integer particle swarm optimization-based task offloading method for optimizing service-level agreement (SLA) satisfaction. *Sang et al. (2022)* proposed a heuristic algorithm for task offloading problem in device-edge-cloud cooperative computing environments with the same objective to our work. These works assumed that all services can be provided by an edge, which does not match with the actual. In real world, each edge has very limited storage capacity, and thus only a few services can be deployed on an edge. Thus, for an efficient task offloading algorithm, the service caching problem must be concerned for deciding which services are deployed on each edge when making offloading decisions.

Focusing on the improvement of task processing performance in edge-cloud computing, several works studied on solving the service caching problem. *Wei et al. (2020, 2021)* proposed a popularity-based caching method based on content similarity. They first produced the popularity of contents based on the historical information of user requests and content similarity, and then replaced most unpopular contents by most popular contents when there is no available storage space in edges. To simplify the edge-cloud system, they assumed that all dynamic parameters were following Poisson distribution in

both their model and simulated experiments. This can lead to an unguaranteed performance in practice and unreliable experiment results. *Wang et al. (2020a)* exploited Markov chain to predict the cached data to improve the hit ratio. *Xia et al. (2021c)* studied on the data caching problem for optimizing the overall service overhead. They modelled the problem into a time-averaged optimization. To solve the problem, they first transformed the optimization model to a linear convex optimization, and then applied Lyapunov optimization technology. *Xia et al. (2021b)* proposed an approximation algorithm for edge data caching (CEDC-A). CEDC-A first choose a solution of caching a data on an ES, with maximum benefit, and then iteratively caching the data with maximum benefit on the ES. In these above two works, the network latency is quantified by the number of hops. While in practice, the real latency is not positively associated with the hop number. All of above works aimed at optimizing the data access latency by designing edge caching strategy. While the task processing performance is not only decided by the data access latency, but also by computing efficiency. Thus, task offloading must be concerned, complementary to edge caching.

*Bi, Huang & Zhang (2020)* studied on the joint service caching and computation offloading for a mobile user allocated to an edge server. They modelled the problem into a mixed integer non-linear programming, and transformed it into a binary integer linear programming. To solve the problem with a reduced complexity, they designed an alternating minimization technique by exploiting underlying structures of caching causality and task dependency. To optimize the overall utility, *Ko et al. (2022)* modelled the processing of user requests as a stochastic process, and proposed to apply water filling and channel inversion for the network resource allocation for users with same service preference. Then they applied the Lagrange multiplier to iteratively solve computation offloading and service caching problem for users with heterogeneous preference. In this work, an edge is assumed to have only one time division multiplexing communication channel. *Tian et al. (2021)* modelled the service caching problem as an MDP, and solved the MDP by combining double deep Q-network (DQN) and dueling-DQN. *Xia et al. (2022)* proposed a two-phase game-theoretic algorithm to provide a strategy of data caching and task offloading. In the first phase, to serve the most users, they iteratively solved the data caching strategy with corresponded user and channel power allocation strategy. In the second phase, they tried to allocate more channel power for each user, for optimizing the overall data transfer rate. These above works didn't concern the allocation of computing resources. *Zhang et al. (2021a)* focused on the joint problem of service caching, computation Offloading and resource allocation, for optimizing the weight cost of computing and network latency. They formulated the problem as a quadratically constrained quadratic program, and used semidefinite relaxation for addressing the problem.

All of these above existing works consider the use of cloud resources only when edge resources are exhausted for offloading tasks. This can lead to an underutilized abundance of cloud resources. Thus, our work tries to exploit the heterogeneity of edges and clouds to improve the cooperation between them, and aiming at providing a joint strategy of service

caching, computing offloading, computing resource allocation, and communication channel assignment, for user satisfaction and resource efficiency optimizations.

## CONCLUSION

In this work, we studied the joint service caching and task offloading problem for edge-cloud computing. We first formulated the problem into a MINLP with objectives of maximizing user satisfaction and resource efficiency, which was proofed as NP-hard. Then, we proposed a polynomial time algorithm with three stages to solve the problem. The basis of our performance was first exploiting abundant cloud resources and low-latency edge resources to satisfy as many requirements as possible in the first two stages, respectively. Then, edge resources were fully used for improving the overall performance in the last stage by re-offloading tasks from the cloud to edges. Simulated experiments are conducted, and the results verified that our method has better performance in user satisfaction, resource efficiency and processing efficiency, compared with five classical and up-to-date methods.

In this article, we focused on the decisions of both task offloading and service caching, without considering the caching replacement. In practice, our method can be applied in collaboration with existing caching replacement approaches. The collaboration efficiency should be studied, which is a focus of our future work. In addition, we will also try to design efficient caching replacement strategy based on predicting user preferences and user similarity, to improve the resource efficiency and processing performance for edge-cloud computing systems.

### Funding

The research was supported by the Key Scientific and Technological Projects of Henan Province (Grant No. 222102210218, 202102210174, 112102210096, 212102210104), the Key Scientific Research Projects of Henan Higher School (Grant No. 20B520039, 21A520050), the National Natural Science Foundation of China (Grant No. 61902021, 61872043, 61975187, 62072414), the Doctoral Fund of Pingdingshan University (No. PXY-BSQD-2022004), the Qin Xin Talents Cultivation Program, Beijing Information Science and Technology University (No. QXTCP B201904), and the fund of the Beijing Key Laboratory of Internet Culture and Digital Dissemination Research (Grant No. ICDDXN004).

### Grant Disclosures

The following grant information was disclosed by the authors:
Key Scientific and Technological Projects: 222102210218, 202102210174, 212102210096 and 212102210104.
Key Scientific Research Projects of Henan Higher School: 20B520039 and 21A520050.
National Natural Science Foundation: 61902021, 61872043, 61975187 and 62072414.
Pingdingshan University: PXY-BSQD-2022004.

Beijing Information Science and Technology University: QXTCP B201904.
Beijing Key Laboratory of Internet Culture and Digital Dissemination Research:
ICDDXN004.

## Competing Interests
The authors declare that they have no competing interests.

## Author Contributions
- Xiaoqian Chen analyzed the data, performed the computation work, prepared figures and/or tables, authored or reviewed drafts of the article, and approved the final draft.
- Tieliang Gao analyzed the data, authored or reviewed drafts of the article, and approved the final draft.
- Hui Gao performed the computation work, authored or reviewed drafts of the article, and approved the final draft.
- Baoju Liu conceived and designed the experiments, performed the experiments, authored or reviewed drafts of the article, and approved the final draft.
- Ming Chen conceived and designed the experiments, performed the experiments, authored or reviewed drafts of the article, and approved the final draft.
- Bo Wang conceived and designed the experiments, performed the experiments, performed the computation work, prepared figures and/or tables, and approved the final draft.

## Data Availability
The resource code implementing the methods evaluated in experiments in C programming language is available in the Supplemental Files.

## Supplemental Information
Supplemental information for this article can be found online at http://dx.doi.org/10.7717/peerj-cs.1012#supplemental-information.

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
