# Peer review of "A multi-stage heuristic method for service caching and task offloading to improve the cooperation between edge and cloud computing"

_PeerJ Computer Science, doi:10.7717/peerj-cs.1012_

## Round 0.1 · original submission · Major Revisions

The paper is very good and there is contribution in this article, however many things must be clarified; different variables with multiple letters must be explained very well, and tables, figures, and the analysis on the structure of MSHCO should also be better explained. The language must also be improved.

Reviewer 1 ·

Basic reporting

The variables need be well defined:
1. The authors defined different variables with multiple letters, i.e., in_i, map_{i,j}, bw_j, nft_i, etc, which are confusing. Commonly, ‘in_i’ in an equation means ‘i’ * ‘n_i’. Therefore, try to define a variable with a single letter.
2. There are multiple variables used in this manuscript. Adding a notation table to list all symbols/notations could improve its readability.
3. There is a highly related work [R1] on service caching and task offloading in MEC, which should be well cited.
[R1] "Joint Optimization of Service Caching Placement and Computation Offloading in Mobile Edge Computing Systems," IEEE Transactions on Wireless Communications, vol. 19, no. 7, pp. 4947-4963, July 2020.

Experimental design

The performance evaluation of the proposed algorithm is not sufficient.
4. With respect to the contribution of this work, since the proposed algorithm MSHCO is a heuristic algorithm, what is the performance gap between MSHCO and the global optimal algorithm? The numerical studies only show that MSHCO outperforms some existing benchmarks. However, we have no idea how good enough is the algorithm, which is essential for potential readers who want to continue studying this topic. Better provide an upper bound or the exhaustive search optimal.
5. This work considers head deadline requirements of tasks, where each task is associated with a deadline d_i. How do you handle the tasks if the deadline cannot be met? Will these unfinished tasks resources for the proposed MSHCO? How about other benchmark algorithms evaluated in the simulations?
6. In the performance evaluation, besides the comparisons with other benchmarks, the analysis on the structure of MSHCO is necessary. For example, how do these three stages of the algorithm affect its performance?

Validity of the findings

no comment

Reviewer 2 ·

Basic reporting

Correct the punctuation error in line 37, used comma two times

On line 56, grammar error “But these works have some issues must be address”

The sentence on line 69, 70, 71, there is single sentence which should be avoided.

In Line 87, “The section” should be replaced with “The next section” or “The second section”

Line 96 have also grammar problem, should be rectified

Please define the sentence on line 199, “and exams the next task”

Another grammar mistake on line 199-200-201 “Otherwise, MSHCO tries to rented a new CS”. “MSHCO rents a new CS with found type”

I think whole paragraph from line 196-205 should be revised because of grammar problems.

Another grammar issue on line 208-209 “There are two situations when offloading a task to an ES, the requested service has or 209 not cached on the ES”

Please also revise sentence on line 217-218.

There are no snapshots regarding experimental work, moreover authors should compare their results in tabular format.

Please also elaborate the comparison of user satisfaction in more detail.

Experimental design

In this paper, the authors have proposed a method to address the joint service caching and task offloading problem in edge-cloud computing environments, to improve the cooperation between edge and cloud resources. They have proposed the method in three steps including formulating the problem into a mix-integer nonLinear programming followed by proposing a a three-stage heuristic method for solving the problem in polynomial time. Finally , they focused on improving the performance of tasks offloaded to the cloud, by re-offloading some tasks from cloud resources to edge resources.
Following are my major questions
It seems to be a stand alone study without the comparison with any state of the art, please try to justify your method with a concrete comparison with some similar kind of latest study. The same should also be discussed in the abstract.

Please briefly justify the potential benefits of the proposed system.
Please make sure to cite the references for the equations if they are not owned by you
What are the formation basis and correctness justifications of Algorithm 2 & 3 ?

Because the cloud & edge computing is the hot area of research these days and there are a lots of test beds available, therefore, authors are strongly recommended to perform a real study instead of the simulations of proposed model. This way , they will have a better chance to prove the authenticity of their work.

Validity of the findings

It seems to be a stand alone study without the comparison with any state of the art, please try to justify your method with a concrete comparison with some similar kind of latest study. The same should also be discussed in the abstract.

What are the formation basis and correctness justifications of Algorithm 2 & 3 ?

Because the cloud & edge computing is the hot area of research these days and there are a lots of test beds available, therefore, authors are strongly recommended to perform a real study instead of the simulations of proposed model. This way , they will have a better chance to prove the authenticity of their work.

Additional comments

please make sure to remove the language and grammatical errors through out the manuscript.

---

## Round 0.2 · accepted · Accept

The authors have responded to all the comments provided by the reviewers. All the comments have been highlighted in the paper, so the paper is in very good standard to be published in PeerJ Computer Science.